# Theoretical Explanation for the Rarity of Antibody-Dependent Enhancement of Infection (ADE) in COVID-19

**DOI:** 10.3390/ijms231911364

**Published:** 2022-09-26

**Authors:** Anna E. Boldova, Julia D. Korobkin, Yury D. Nechipurenko, Anastasia N. Sveshnikova

**Affiliations:** 1Center for Theoretical Problems of Physico-Chemical Pharmacology, Russian Academy of Sciences, 30 Srednyaya Kalitnikovskaya Str., 109029 Moscow, Russia; 2Engelhardt Institute of Molecular Biology, Russian Academy of Sciences, 119991 Moscow, Russia; 3Department of Normal Physiology, Sechenov First Moscow State Medical University, 8/2 Trubetskaya St., 119991 Moscow, Russia; 4Faculty of Fundamental Physico-Chemical Engineering, Lomonosov Moscow State University, 1/51 Leninskie Gory, 119991 Moscow, Russia

**Keywords:** COVID-19, antibody-dependent enhancement, computational modeling, SARS-CoV-2

## Abstract

Global vaccination against the SARS-CoV-2 virus has proved to be highly effective. However, the possibility of antibody-dependent enhancement of infection (ADE) upon vaccination remains underinvestigated. Here, we aimed to theoretically determine conditions for the occurrence of ADE in COVID-19. We developed a series of mathematical models of antibody response: model Ab—a model of antibody formation; model Cv—a model of infection spread in the body; and a complete model, which combines the two others. The models describe experimental data on SARS-CoV and SARS-CoV-2 infections in humans and cell cultures, including viral load dynamics, seroconversion times and antibody concentration kinetics. The modelling revealed that a significant proportion of macrophages can become infected only if they bind antibodies with high probability. Thus, a high probability of macrophage infection and a sufficient amount of pre-existing antibodies are necessary for the development of ADE in SARS-CoV-2 infection. However, from the point of view of the dynamics of pneumocyte infection, the two cases where the body has a high concentration of preexisting antibodies and a high probability of macrophage infection and where there is a low concentration of antibodies in the body and no macrophage infection are indistinguishable. This conclusion could explain the lack of confirmed ADE cases for COVID-19.

## 1. Introduction

Since 2019, coronavirus disease (COVID-19) has spread all over the world. Severe forms of COVID-19 are associated with acute respiratory distress syndrome, pneumonia, renal failure and death [1,2]. Global vaccination is a path to the reduction of the disease’s severity, as well as its spreading rate [3]. Nevertheless, among the general concerns for the development and utilization of vaccines is the possibility of antibody-dependent enhancement of infection (ADE) [4]. Besides vaccination, ADE is frequently caused by viruses that have high antigenic diversity and the ability to replicate in immune cells [5], such as coronaviruses, Ebola virus, dengue virus, human immunodeficiency viruses (HIV) and influenza [6,7,8,9,10]. ADE mostly occurs when a non-neutralizing or poorly neutralizing antibody (Ab) binds to a virus particle [11]. Normally, binding of the Ab to the Fc receptors (FcRs) on leukocytes leads to the destruction of the virus inside these cells [12]. However, in the case of ADE, Abs can release the virus and let it replicate in FcR-expressing cells [13]. Moreover, such infection of immune cells can cause adverse responses [11], such as enhanced cytokine release by the infected cells, which facilitate tissue damage [14].

ADE can originate from different features of (pre-existing) antibodies. For dengue infection, it was shown that both non-neutralizing antibodies and antibodies that are properly neutralizing but possess a low affinity to parts of the virus may be associated with ADE [15]. Specifically, for severe acute respiratory syndrome coronavirus (SARS-CoV-2), it was shown that ADE-causing antibodies, regardless of their antigen affinity, could bind to RBDs in two different states, while the neutralizing mAbs, which had no ADE activity, were only able to bind to RBD in the S trimer in a single state [15]. Another major component of the antiviral activity of IgG antibodies is their capacity to engage and activate specific FcγR pathways. There are numerous instances where high-affinity neutralizing antibodies fail to offer protection in vivo when Fc–FcγR interactions are weak and where antibodies with poor neutralizing activity in in vitro assays provide robust antiviral protection in vivo [16].

Cases of ADE have been reported upon infection by SARS-CoV [4,17]: non-neutralizing anti-spike sera can initiate viral entry into non-ACE-2-expressing cells both in vitro [17] and in vivo [4]. Indeed, specific Abs for the RBD domain of SARS-CoV S-protein can mediate the entry of the virus into FcR-expressing human cells [18]. Furthermore, vaccination-induced neutralizing antibodies potentiated infection of B-cell cell culture by SARS-CoV in an FcR-dependent manner [17]. The same phenomenon has been observed for the Middle East respiratory syndrome coronavirus (MERS-CoV), where ADE was facilitated in cases of low antibody titers, while high Abs neutralized the virus [19]. It is noteworthy that, in SARS-CoV-infected macaques, non-neutralizing antibodies against the S protein were associated with fatal acute lung injury attributed to alterations in pro-inflammatory immune responses [20].

Whether ADE could contribute to COVID-19 clinical pathogenesis is controversial [21,22]. On the one hand, there are almost no confirmed clinical cases of ADE in COVID-19; on the other hand, there are data on SARS-CoV-2 replication in macrophages [23,24,25], which lead to ADE. It should be noted that in vitro SARS-CoV-2 does not penetrate primary peripheral blood mononuclear cells [23]. In some studies, SARS-CoV-2 replication in immune cells has been shown to be abortive [24,25], as no viable virions were produced [25,26,27], or statistically insignificant [28]. Nevertheless, COVID-19 patient autopsy analysis revealed the presence of SARS-CoV-2 RNA not only in the localized granules but also with large cytoplasmic volumes in macrophages [29], which can indicate possible viral replication. Moreover, other coronaviruses can use monocytes and macrophages as hideouts [30] and, thus, be distributed throughout the organism from the lungs [31]. Despite the number of published [4,22,32] and pre-printed studies [33] discussing the possibility of ADE in COVID-19, there are no clinical studies confirming ADE. 

In this study, we constructed a set of mathematical models capable of mechanistic description of complex interactions between the virus, lung cells, Abs, B-cells and macrophages. Analysis of the models demonstrated that ADE can indeed occur in COVID-19. The affinity of the antigen–antibody complex influenced the course of the disease; however, its contribution depended on the level of antibodies and the macrophage infection probability. The required level of the pre-existing antibodies should be high enough for ADE observation. Under conditions of macrophage infection, pneumocyte survival is no lower than in the absence of ADE because of the neutralizing activity of high antibodies doses. These results may explain the lack of reported ADE cases in COVID-19 patients.

## 2. Results

### 2.1. Construction of Antibody Response Generation Module (Model Ab)

In order to describe antibody response to the SARS-CoV-2 infection, we constructed the first model, which takes into account the interaction between virus particles and the antibody-production system (B-cells). In this model, first, the average dynamics of viral load [34] were approximated phenomenologically with a function described by Equation (1) (see Section 4) (Figure 1a), and then the model was adjusted to the experimental data for the antibody response of each patient by variation of five model parameters: the parameter *β_RNA_*, which describes average number of antigens per viral RNA molecule; the parameter *w_3_*, which describes individual viral load; and the intrinsic parameters of the immune system (time delay for naive B-cell activation *t_naive_*; plasma cells death rate *δ_pl_*; and B-cell differentiation rate, *π_pl_*) (Appendix A). A description of a typical individual patient’s antibody response [34] is given in Figure 1b. The kinetics of free viral epitopes for different *β_RNA_* and *w*_3_ are shown in Appendix A. 

The following average values of parameters for the given group of patients (6 patients with mild and 7 patients with severe course of disease) were obtained: *β_RNA_* = 532.38, *w*_3_ = 2.378 days, *t_naive_* = 10.5 days, *δ_pl_* = 0.166 days^−1^, and *π_pl_* = 10^−4^ (APS∙days)^−1^. Typical dynamics of naive B-cells, plasma cells and memory B-cells are given in Figure 1c, Appendix A. Altogether, model Ab correctly describes antibody response, B-cell differentiation and immune system-mediated virus elimination in response to the given viral load. However, this model does not take into account variabilities in the types of antibodies and the possible infection of macrophages. 

### 2.2. The Infection Propagation Model (Model Cv) Suggests Possibilities of ADE Occurrence in COVID-19

The infection propagation model (model Cv) was developed to describe antibody-dependent infection of macrophages. The model parameters were adjusted based on viral replication kinetics in HAE cells [35] (Appendix A) and the viral load data from the sputum of seven COVID-19 patients from Germany [1] (Appendix A and Figure 1d). The model was able to describe viral load data for different patients upon variation of such parameters as pneumocyte infection rate β_inf_, the maximum virus–antibody complex formation constant α_c0_, the rate of viral degradation δ_v_ and the initial viral load *Virus*_0_. After approximation of seven experiments, the following average parameter values were obtained: β_inf_ = 4.72 × 10^−6^ (days∙Virus)^−1^, α_c0_ = 5.99 × 10^−4^ days^−1^, δ_v_ = 207.6 days^−1^ and *Virus*_0_ = 2.6 × 10^7^. Other parameters are given in Appendix A. The dynamics of the viral load and the pneumocyte number for different ρ are given in Figure 1e,f.

We define ADE as a drastic drop in the concentration of healthy pneumocytes (more than 5% in comparison with the course of disease without macrophage infection). Model Cv could predict conditions for the occurrence of ADE for average antibody level. For this, we calculated the dependence of the viral load dynamics on the probability of macrophage infection (ρ) (Figure 1e). At ρ < 10^−3^, the concentration of viral particles was the same as in the absence of macrophage infection, and no ADE was observed, while for ρ > 10^−3^, the probability of pneumocyte survival decreased and the times of viral clearance increased, allowing the occurrence of ADE. However, we could not investigate the impact of antibody parameters on ADE occurrence using only model Cv because the existing experimental data describe antibodies in terms of optical density and we needed module model Ab to obtain the absolute concentrations of antibodies.

### 2.3. The Complete Model Is Capable of Describing Viral Load and Assessing the Influence of Pre-Existing Antibody Concentration on the Course of Disease

Model Ab described the antibody dynamics and model Cv predicted possible ADE occurrence. The combined model (complete model) included not only the dynamics of antibodies and the formation of immune memory but also the death of pneumocytes and the possible infection of macrophages after absorption of the antigen–antibody complex. The details of the validation of the complete model are given in the Materials and Section 4 and in Appendix A. 

The complete model described both the viral load dynamics and the antibody response. Within the framework of this model, we estimated the initial viral load (*V*_0_) impact on the viral load dynamics (Figure 2a) and on the percentage of infected pneumocytes (Figure 2b). The increase in the initial antigen concentration led to the earlier appearance of the viral load peak and a slight increase in its magnitude (Figure 2a). The amount of surviving pneumocytes at the end of the disease decreased with increasing initial viral load (Figure 2b).

Then, we estimated the impact of the concentration of pre-existing antibodies (Figure 2c,d). The model predicted that only the highest concentration (*A*_0_ ≈ 10^13^ particles) of antibodies could significantly change the course of disease; i.e., the number of surviving pneumocytes increased drastically (Figure 2d), while the viral load noticeably decreased (Figure 2c). As discussed above, the dissociation constant (*K_d_*) of the antigen–antibody complex could affect the course of the desease. To determine the extent of *K_d_* influence, we performed similar simulations for varied values of *K_d_* (Figure 2e–h). In the absence of pre-existing antibodies, Kd did not influence the viral load maximum, but larger Kd led to lower pneumocyte survival (Figure 2f) and prolonged virus elimination (Figure 2e). In the presence of pre-existing antibodies at the maximum concentration (*A*_0_ ≈ 10^13^ particles), the viral load decreased with time, all pneumocytes survived for small Kd values (Kd < 10^−9^ M) and up to 20% of the pneumocytes died for large Kd values (Kd > 10^−9^ M). Therefore, the existance of low-affinity pre-existing antibodies could lead to ADE. 

### 2.4. Theoretical Conditions for ADE Occurrence in COVID-19

The next question of interest was whether ADE could occur in primary infection. To answer this question, we calculated the number of pneumocytes, the viral load and the number of life macrophages in the absence of pre-existing antibodies (Figure 3). Our results showed that ADE could occur in the absence of pre-existing antibodies if the probability of macrophage infection was greater than 10^−3^. At ρ < 10^−3^, the concentration of viral particles was the same as in absence of macrophage infection. Upon an increase in the probability of macrophage infection, pneumocyte survival decreased and viral load clearance time increased.

In order to determine the sets of conditions in which ADE could occur in the course of SARS-CoV-2 infection, we performed complete model simulations for sets of parameters with varied pre-existing antibody concentrations, dissociation constants for the antigen–pre-existing antibody complex and probabilities of macrophage infection (Figure 3, Figure 4 and Figure 5). 

First of all, we investigated the influence of pre-existing antibody concentration on the course of disease (Figure 4 and Appendix A) for a given probability of macropage infection (*ρ*). For *ρ* < 3 × 10^−4^, ADE was not observed (the percentage of dead pneumocytes did not exceed levels 3% higher than with the course of disease without macrophage infection; this situation occurred with extremely high antibody concentrations 10^14^ < *Ab_old_* < 10^15^). For ρ > 3 × 10^−4^, ADE occurred only if the concentration of pre-existing antibodis was high enough (*Ab_old_* > 10^10^). 

Secondly, we estimated the role of the antigen–pre-existing antibody complex dissociation constant (Kd) in possible ADE occurrence (Figure 5 and Appendix A). Interestingly, the influence of Kd on the course of disease was depended on the macrophage infection probability ρ (see Appendix A for details). For ρ < 3 × 10^−4^, the proportion of surviving pneumocytes increased upon a decrease in Kd. In this case, the maximal number of dead pneumocytes was no greater than their amount in the absence of antibodies and possible macrophage infection (Figure 5). For ρ > 3 × 10^−4^*,* the highest level of pneumocyte death was observed at the lowest Kd. In all these cases, the degree of Kd impact on pneumocyte survival depended on the pre-existing antibody concentration (Figure 5). 

## 3. Discussion

The goal of our study was to theoretically determine the conditions for ADE occurrence in COVID-19. 

First of all, we constructed an antibody response generation model that could describe antibody dynamics in COVID-19. We modified the mathematical model [36] for ADE in Dengue fever using parameters corresponding to SARS-CoV-2 infection to describe the observed viral load kinetics. The constructed model was able to describe the experimental viral load dynamics [1], seroconversion times and antibody concentrations [34]. We analyzed the influence of key parameters on the course of the disease (Figure 2). We found that an increase in the initial viral load induced an increase in the maximum viral load and caused an earlier rise and decline in the viral load. This result corresponds with the predictions of other SARS-CoV-2 mathematical models [37]. For other type of virus (influenza), an increase in the initial viral load led to a time-shift of the peak without changing its magnitude [38]. This difference could be explained by differences in the assumed host cell behavior; specifically, in our model, there was no mechanism for pneumocyte recovery after infection, unlike other mathematical models for respiratory infections [38,39]. 

In this study, we assumed that ADE occurs if the amount of dead pneumocytes is higher in the presence of pre-existing antibodies. It should be noted that, in contrast to previously published models [36], in the mathematical model constructed ere the pneumocyte damage varied from patient to patient (10–90%) and never reached 100%. 

Using the complete model, we found that the probability of macrophage infection regulated the impact of antibody concentration over the course of disease (Figure 4 and Figure 5). At any probability of macrophage infection, the concentration of pre-existent antibodies played a key role in antibody-mediated enhancement of disease. This result corresponded with a previously predicted high impact of antibody concentration on ADE occurance [39], although in that work no explicit immune cells were included. Our calculations demonstrated that, with moderate macropage infection probabilities, ADE was possible only for the highest antibody concentrations (Figure 4 and Appendix A). Similar results were obtained experimentally for dengue disease [40]. The lower the concentration of antibodies in the body was, the higher the probability of macrophage infection required in order to cause significant changes in the kinetics of the viral load and survival of pneumocytes. When the concentration of antibodies was high, the proportion of infected macrophages increased. However, in such cases, antibodies began to have a neutralizing effect. As a result, the number of dead pneumocytes in cases with possible infections of macrophages and a high antibody titer was no greater than the number of dead pneumocytes in cases where the level of antibodies was low and there was no antibody-mediated infection of macrophages. Thus, even if an antibody-dependent macrophage infection is detected, it is most likely that the lungs will not be more affected than in the course of the disease in the absence of ADE. 

The antigen–antibody affinity can affect the initiation of ADE, but its influence on the course of the disease is controlled by the level of antibodies in the body. A marked decline in pneumocyte survival will only be observable when the binding affinity between the viral epitope and the antibody is very high. This suggestion is indirectly supported by a clinical case in which an acute myeloid leukemia patient might have suffered ADE shortly after being treated with high-affinity anti-COVID-19 therapy [41].

However, a situation in which the antigen–antibody affinity for the newly synthesised antibodies is lower than for pre-existing antibodies seems extremely unlikely, significantly lowering the probability of ADE after vaccination. Indeed, consistent with [42,43], it can now be assumed that anti-SARS-CoV-2 vaccination does not aggravate the disease course in COVID-19.

There is a set of experimental findings that were not taken into account by the developed model and which could have possibly made it underestimate the effect that non-neutralizing antibodies have on the course of disease. First, it is noteworthy that lung epithelial cells also express high levels of FcγRIIa, and non-neutralizing antibodies might facilitate their infection [44]. In addition, COVID-19 patients have been reported to have a strong IgG antibody response to the nucleocapsid protein, which cannot neutralize SARS-CoV-2 at all, resulting in delays in virus clearance and an increased severity of infection [44].

Another point to be discussed is the development of ADE upon treatment with convalescent plasma. Although the mathematical model proposed here cannot be directly applied in this case, some data can be obtained from such studies. Altogether, the model results are in accordance with the data from [45], where it was shown that, in a cell culture, donor and patient plasma could induce ADE, but its probability did not differ between plasma samples from convalescent, mild, moderate and severe patients. In contrast to this data, a study on the transfer of neutralization activity with COVID-19 convalescent transfusion [46] demonstrated the absence of ADE upon transfusion. Based on our calculations, we speculate that convalescent plasma transfer is not likely to change the course of disease, but other plasma transfusion risks, such as hemolytic transfusion reactions, anaphylactic reactions, transfusion-related acute lung injury and allergic reactions, might occur [47].

## 4. Materials and Methods

### 4.1. General Principles for the Construction of the Mathematical Models 

The models contained several compartments, with the assumption of well-mixing of species inside each compartment (homogeneous models). The models were based on ordinary differential equations derived from the laws of chemical kinetics (the mass action law, Michaelis–Menten kinetics or Hill equation). The schemes of the models are given in Figure 1. Model parameters were taken from the literature data or estimated based on the available experimental data, as explained in the Supporting Information. 

In order to describe the ADE phenomenon through mathematical modeling, we defined ADE in the text as a drastic drop in the concentration of healthy pneumocytes (more than 5% in comparison to the course of disease without macrophage infection).

### 4.2. Model Ab—The Antibody Response

Model Ab was constructed to describe interactions between the pathogen (a virus particle) and the antibody-production system (Figure 6a). It included induction of Ab synthesis in lymphoid organs, binding of the Abs to viral particles and Ab-dependent degradation of viruses. Model Ab comprised two different compartments: lungs and lymphoid organs. B-cell activation, proliferations, differentiation and antibody-production occur in lymphoid organs, whereas viral interaction with antigen-presenting cells and antibodies takes place in the lungs. We assumed that the virus–antibody complex formed between a single viral protein (epitope) and an antibody independently of steric interference between the proteins on the viral surface. 

In model Ab, it was assumed that the number viral particles (Virus) was proportional to the number of SARS-CoV-2 RNA in patient samples. The kinetics of the SARS-CoV-2 RNA number in saliva was set analytically as:(1)$$F_{RNA}=A_{RNA}*\frac{1}{1+e^{-\frac{t-x_{0}+\frac{w_{1}}{2}}{w_{2}}}}*(1-\frac{1}{1+e^{-\frac{t-x_{0}-\frac{w_{1}}{2}}{w_{3}}}}),$$
where *F_RNA_* is the concentration of SARS-CoV-2 RNA, t is the model time and *A_RNA_*, *x*_0_, *w*_1_, *w*_2_ and *w*_3_ are parameters required to better fit the experimental data [34]. Numerical values of these parameters are represented in Appendix A. It was assumed that the parameter *w*_3_ was specific for individual patients. Individual parameter values are represented in Appendix A.

In the frame of model Ab, we assumed that each SARS-CoV-2 virus particle had exactly 25 S proteins on its surface and each protein on the virus could bind antibodies independently [48]. Thus, the number of viral particles was proportional to the sum of the viral epitope bound with the antibody (*Va*) and the free viral epitope (*V_f_*). Subsequently, the number of free viral epitope could be described with the following formula:(2)$$V_{f}=\beta_{RNA}*F_{RNA}-Va,$$
where *β_RNA_* is the constant of proportionality between the SARS-CoV-2 RNA number and the amount of free viral epitopes. It was assumed that this parameter implied the specificity of the viral kinetics in individual patients. Individual parameter values are represented in Appendix A.

In model Ab, it was assumed that the clearance rate of the epitope–antibody complex was approximated by the Michaelis–Menten formula, as described in the following equation:(3)$$\frac{dVa}{dt}=k_{v1}*V_{f}*Ab_{lung}-k_{v-1}Va-\frac{C_{v1}}{C_{v2}+Va}Va$$
where *Ab_lung_* is the concentration of antibodies in the lungs; *k_v_*_1_ and *k_v_*_−1_ are the epitope–antibody association and dissociation rates, respectively; *c_v_*_1_ is the maximal rate of clearance of viral protein–antibody complexes; and *c_v_*_2_ is the half-decay constant.

The next equations for the model describe the Ab production in the response to a viral load. Formation of the long-term immune response is a complex process governed by multiple factors, such as the number of antigen-presenting cells, T-cell help, the area of contact between T- and B-cells on the border of lymph nodes, etc. [48]. In order to avoid excessive complexity, we unified all of these factors through the introduction of antigen-presenting sites (APSs). APSs are represented for the lungs and lymphoid organs. Lung APSs can exist in two states: neutral (*Ap*) and activated (*APS_lung_*). Activation of APSs occurred in the lungs when viral epitopes interacted with antigen-presenting cells. B-cells could be activated in the lymphoid organs. Transition of activated antigen-presenting sites in the lung (*APS_lung_*) into activated antigen-presenting sites in lymphoid organs (*APS_lo_*) occurred in accordance with the law of mass action. In the absence of a pathogen, all sites were assumed to be in a neutral state, *Ap*, and they could not affect B-cell functioning. The dynamics of neutral APSs and activated APSs in lungs and lymphoid organs can be described by the following equations: (4)$$\frac{dAp}{dt}=\chi_{ap}\left(Ap_{0}-Ap\right)-Ap*\frac{\beta_{apm}*V_{f}}{\varphi +V_{f}}$$
(5)$$\frac{dAps_{lung}}{dt}=Ap*\frac{\beta_{apm}*V_{f}}{\varphi +V_{f}}-\nu \left(k_{s1}*Aps_{lung}-k_{s-1}*Aps_{l0}\right)-\delta_{ap}Aps_{lung}$$
(6)$$\frac{dAps_{lo}}{dt}=k_{s1}*Aps_{lung}-k_{s-1}*Aps_{l0}-\delta_{apm}Aps_{lo}$$
where χ_b_ is the generation and degradation of neutral APSs in the absence of the virus (homeostasis rate of neutral APSs); *Ap*_0_ and *Ap* are the initial and transient concentrations of neutral APSs; *δ_ap_* and *δ_apm_* are the rates of *APS_lung_* and *APS_lo_* deactivation, correspondingly; the terms *β_apm_* ∗ *Ap* ∗ *V_f_*/(*φ* + *V_f_*) indicate the rate of *APS_lung_* activation, *β_apm_* is the maximal activation rate; and *φ* is the half-saturation constant for *APS_lung_* activation. *φ* was estimated from an existing model [49,50]. The terms *k_s_*_1_ and *k_s_*_−1_ denote rates of APS transition between two compartments: lungs and lymphoid organs. *ν* is the conversion coefficient reflecting the lymphoid organ:lung volume ratio.

Activation of B-cells was assumed to occur in a separate compartment in lymphoid organs, such as the spleen or lymph nodes [48]. After interaction with activated sites (*APS_lo_*), naive B-cells (B) proliferated and differentiated into short-lived IgM-producing plasma cells, long-lived IgG-producing plasma cells (Pl) and memory B-cells (Bm) (Figure 5a). We focused on B-cells, which produce high-affinity class-switching IgG-antibodies. The possibility of B-cell differentiation into short-lived IgM-producing plasma cells was also included in the model. In the frame of the model, we assumed that B-cell proliferation and differentiation into short-lived plasma cells occurred after their contact with APSs, whereas differentiation into long-lived plasma cells and memory B-cells occurred after a time delay. The complete equation for the concentration of B-cells is the following: (7)$$\frac{dB}{dt}=\chi_{b}(B_{0}-B)+\pi_{b}Aps_{lo}B-Aps_{lo}B\left(\tau_{n}*\pi_{pl}+\tau_{n}*\pi_{bm}+\pi_{ps}\right)$$
$$\tau_{n}=\{1,\;if\;\left(t-t_{0}\right)>t_{naive}\;0,\;if\;\left(t-t_{0}\right)\leq t_{naive}\;$$
where *B*_0_ and *B* are the initial and transient concentrations of B-cells; χ*_b_* is the B-cell generation and degradation in the absence of the virus (homeostasis rate of B-cells); *π_b_* is the naive B-cell proliferation rate (reflecting the increase in B-cell proliferation upon contact with APSs) [51]; the *π* parameters reflect probabilities of different B-cell fates upon contact with the APSs: *π_pl_*—into plasma cells, *π_bm_*—into memory cells and *π_ps_*—into short-lived Ig-M-producing plasma cells; *τ_n_* is used to introduce a time delay between the appearance of APSs and the initiation of naive B-cell differentiation into long-lived plasma cells and memory B-cells [48]; *t*_0_ is the moment when *APS_lo_* becomes positive; and *t_naive_* is the lag-time required for B-cell differentiation.

The next equation for model Ab described the dynamics of IgG-producing plasma cells (Pl):(8)$$\frac{dPl}{dt}=\tau_{n}\;\pi_{pl}Aps_{lo}B+\tau_{mem}\pi_{ml}Aps_{lo}B_{m}-\delta_{pl}Pl$$
$$\tau_{mem}=\{1,\;if\;\left(t-t_{0}\right)>t_{memory}\;0,\;if\;\left(t-t_{0}\right)\leq t_{memory}$$
where *δ_pl_* is the rate of plasma cell degradation and *τ_mem_* reflects the time lag (*t_memory_*) between the appearance of APSs (at *t*_0_) and the initiation of memory B-cells differentiation and proliferation [48].

As well as the formation from B-cells and production of plasma cells, the memory B-cells were assumed to be capable of reproduction with saturation [48]. Thus, the memory cell dynamics were described by the following equation.
(9)$$\frac{dB_{m}}{dt}=k_{bm1}B_{m}\left(1-\frac{B_{m}}{k_{bm2}}\right)+\tau_{n}\pi_{bm}Aps_{lo}B-\tau_{mem}\;\pi_{ml}Aps_{lo}B_{m}$$
where *k_bm_*_1_ and *k_bm_*_2_ represent the *B_m_* growth rate and the lymphoid organ maximum capacity, respectively.

The concentration of antibodies in the lymphoid organs and body fluids (*A_b_*) could be described with the following equation:(10)$$\frac{dAb_{lo}}{dt}=\alpha_{pl}Pl-ka_{-1}*Ab_{lo}+ka_{1}Ab_{lung}-\delta_{A}Ab_{lo}$$
where *α_pl_* denotes the rate of IgG generation by long-living plasma cells [52] and *δ_a_* is the natural decay rate of antibodies (*A_lo_* and *A_lung_*) [38,51]. The terms *k_a_*_1_ and *k_a_*_−1_ denote rates of antibody transition between two compartments, the lungs and lymphoid organs.
(11)$$\frac{dAb_{lung}}{dt}=-k_{v1}V_{f}*Ab_{lung}+k_{v-1}Va+\nu \left(ka_{-1}*Ab_{lo}-ka_{1}Ab_{lung}\right)-\delta_{A}Ab_{lung}$$

To compare the model predictions with experimental data on serum antibody levels [34], we introduced into the model Ab parameters *ω* and *Ig*0, which indicate the background level and a scaling coefficient that depended on the type of antibody test. Thus, the relative IgG equaled ln(*Ab_lo_/Ig*0*+ω*). 

### 4.3. Model Cv—Propagation of Infection

Model *Cv* was constructed to describe the elimination of viral particles by alveolar macrophages, viral entry and replication in pneumocytes and antibody-mediated viral entry into macrophages (Figure 6b). Model *Cv* described events localized in a lung alveolus; therefore, the parameters differed from model *Ab*. 

The antibodies were assumed to be produced by plasma cells, the concentration of which could be described by a simplified version of Equation (8):(12)$$\frac{dPl_{mod}}{dt}=\kappa_{b}+\alpha_{B}Pl_{mod}*Virus-\mu_{B}Pl_{mod}$$
where *κ_b_* and *μ_B_* are the plasma cell production and degradation rates and *α_B_* is the rate of plasma cell proliferation upon contact with viral particles [36]. As antibodies are produced by plasma cells, in model *Cv* we assumed the concentration of antibodies to be proportional to the concentration of plasma cells and, in the following equations, *Pl_mod_* is used to indicate Ab concentration and *Pl_mod_* * Virus is used to indicate the concentration of virus–antibody complexes. 

In model *Cv*, macrophages were assumed to be in two states: healthy (*M_A_*) and infected (*M**). Healthy macrophages engulfed viral particles without changing their concentration or engulfed virus–antibody complexes and became infected. In the model, it was assumed that the infected macrophages could recover [36] and both healthy and infected macrophages could degrade. With the described assumption, the equations for *M_A_* and *M** were the following:(13)$$\frac{dM_{A}}{dt}=\chi_{M}(M_{0}-M_{A})+\sigma M^{*}-\rho \alpha_{c}Pl_{mod}*Virus*M_{A}$$
(14)$$\frac{dM^{*}}{dt}=\rho \alpha_{c}Pl_{mod}*Virus*M_{A}-\left(\sigma +\delta_{I}\right)M^{*}$$
where *χ_M_* is the macrophage generation and degradation in the absence of a virus (homeostasis rate of macrophages); *δ_I_* is the death rate of infected macrophages (*δ_I_* ≥ *χ_M_*); *σ* is the rate of infected macrophage recovery; *ρ* denotes the probability of macrophages becoming infected; and *α_c_* is the rate for antibody–virus complex formation and subsequent engulfment, expressed as a step function:$$a_{c}=a_{c0}*\frac{t^{5,5}}{t^{5,5}+{\left(t_{ser}\;*\;2\right)}^{5,5}}$$
where *α_c_*_0_ is the maximal constant rate, t is the model time and *t_ser_* is the day of seroconversion. The function of the rate for antibody–virus complex formation and the corresponding parameters were chosen to allow approximation of antibody kinetics with model Ab. We used $t_{ser}*2$ to acquire a better approximation of the antibody kinetics.

Healthy pneumocytes (*P_n_*) were assumed to be the main virus-producing cell. They could engulf the virus, become infected (*P_i_*) and, consequently, die [53]:(15)$$\frac{dP_{n}}{dt}=-\beta_{inf}P_{n}*Virus$$
(16)$$\frac{dP_{i}}{dt}=\beta_{inf}P_{n}*Virus-\delta_{p}P_{i}$$
where *β_inf_* is the rate of infection of pneumocytes and *δ_p_* is the death rate of infected pneumocytes.

Thus, viral concentration was described with the following equation: (17)$$\frac{dVirus}{dt}=pP_{i}-\alpha_{c}Pl_{mod}*Virus*M_{A}-\left(\alpha_{v}M_{A}+\delta_{v}\right)*Virus$$
where *p* is the rate of virus production in pneumocytes, *α_v_* is the rate of viral phagocytosis by the macrophages and *δ_v_* is the rate of macrophage-independent viral clearance.

We assumed that the number of viral particles was proportional to the RNA concentration obtained by PCR in COVID-19 patients. Parameter αRNA was added to connect viral RNA concentration in sputum to the actual virus concentration in the lungs. It was individually chosen simultaneously for all patient data and selected to be in the range from 100 to 0.001. 

### 4.4. Complete Model—A Combination of Model Ab and Model Cv

The complete model was designed to combine the antibody production from model Ab and the production of viruses and infection of macrophages from model *Cv*. The scheme of the model is given in Figure 6c. The required modifications of the models included scaling of the volumes and concentrations, addition of macrophage-independent degradation of both free and antibody-bound virus epitopes, conversion of viral particle concentrations (Viral) from model *Cv* into viral epitope concentration (*V*) from model *Ab* and direct inclusion of the virus–antibody complexes (*Va*) with the equations for *M_A_* and *M**. Therefore, Equations (2), (3), (13) and (14) were modified as follows: (18)$$\frac{dV}{dt}=\pi_{V}^{*}P_{i}-k_{v1}V*Ab_{lung}+k_{v-1}Va-(\alpha_{v}^{}*M_{A}+\frac{C_{v1}^{}}{C_{v2}^{}+V})V$$
(19)$$\frac{dVa}{dt}=k_{v1}V*Ab_{lung}-k_{v-1}Va-\alpha_{c}^{*}*M_{A}*Va$$
(20)$$\frac{dM_{A}}{dt}=\chi_{M}(M_{0}-M_{A})-\rho^{*}\alpha_{c}^{*}{}^{}VaM_{A}+\sigma M^{*}$$
(21)$$\frac{dM^{*}}{dt}=\rho^{*}\alpha_{c}^{*}Va*M_{A}-\left(\sigma +\delta_{I}\right)M^{*}$$

Parameters *π_V_** and *ρ** were recalculated from *p* and ρ by dividing them by the number of S proteins on the viral particle. Here, we did not include viral production by macrophages in Equation (18). In some studies, SARS-CoV-2 replication in immune cells has been shown to be abortive, as no viable virions were produced [25,26,27].

In the complete model, the infection probability was linked to the features of the antibodies indirectly because it is currently assumed that a set of antigen–antibody interaction properties are suspected to cause ADE [15,54], and one of the most important properties of IgG antibodies is their capacity to engage and activate specific FcγR pathways, which does not depend on their affinity [16].

### 4.5. Model Integration, Analysis and Parameter Estimation 

The systems of ordinary differential equations were integrated using the LSODA method [55,56] in COPASI software (http://www.copasi.org, accessed on 1 January 2020). 

Estimations of model parameters were undertaken separately for model Ab and model *Cv* based on experimental data or previously published models, as described below and in the Supporting Information (Appendix A). Additional parameter adjustment for the complete model was based on experimental data [1,35,57].

For model Ab, values for naive B-cell proliferation, antibody secretion by plasma cells and antibody death rates were taken from published mathematical models and the literature [38,51,58], as indicated in Appendix A. Parameters for the antigen–antibody complex formation were obtained from the average values for binding with monoclonal antibodies [59]. The ranges for time lags t_memory_ and t_naive_ were calculated from known data on lymphocyte physiology [60,61] and experimental results [34] (see Appendix A for the calculation details). 

For model *Cv*, parameters were either fitted for personal patient data [1] or taken from [36] (Appendix A).

For the complete model, the parameters *π_V_** and *δ_p_* were estimated from the data on SARS-CoV-2 replication in human airway epithelial (HAE) cells [35]. The set of equations for HAE cell infection is represented in Appendix A. The parameters for the macrophage activity were calculated based on experimental data and published models [1,38,57], as described in Appendix A.

The analysis of the models consisted of time-course simulations, analysis of steady states performed using Newton’s method [55] in COPASI and sensitivity analysis [55,62,63]. For the latter, we calculated sensitivities using the “one-at-a-time” approach and the following formula:(22)$$S_{j}^{i}=\frac{\Delta lnln\;R_{i}\;}{\Delta lnln\;p_{j}\;}$$
where $R_{i}$ indicates the model response (*Va*, *S_a_*, *A_b_*, *B*, *Lp*, *Bm* at time 30 days for model Ab) and *p_j_*-s are the models parameters. The detailed results are represented in Appendix A.

## Figures and Tables

**Figure 1 ijms-23-11364-f001:**
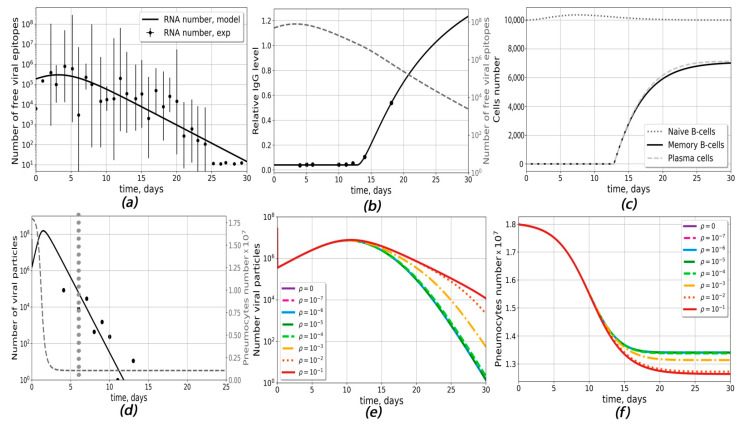
Antibody-response generation (model Ab) and infection propagation (model Cv) outputs. (**a**–**c**) model Ab: (**a**) approximation of experimental data for viral load dynamics [34]. The following parameter values were chosen (Equation (1)): A_RNA_ = 798,292 RNA numbers, x_0_= 2.998 days, w_1_ = w_2_ = 2.069 days^−1^, w_3_ = 2.38 days. (**b**) Changes in relative numbers of antibodies and dynamic of viral epitopes for an individual patient [34]. (**c**) Kinetics of naive B-cells, plasma cells and memory B-cells for the same patient. The following donor-specific parameters were used: δ_pl_ = 0.004 days^−1^, π_pl_ = 3 × 10^−5^ (APS∙days)^−1^, *β_RNA_* = 264, w_3_ = 2.62 days, t_naive_ = 12.8 days, w = 1.04. (**d**–**f**) model Cv (**d**) A typical fit to the experimental data (black dots) on viral load in COVID-19 patients [1]. The day of seroconversion is marked with the grey dotted line and assumed to happen on day 6. The following donor-specific parameters were used: β_inf_ = 2.21 × 10^−6^ (Virus∙days)^−1^, δ_v_ = 255 days^−1^, ac_0_ = 2.17 × 10^−10^ (cells∙cells∙days)^−1^, Virus_0_ = 1.32 × 10^7^ particles; (**e**) The dependence of the viral response dynamics for different values of the probability on macrophage infection probability ρ. (**f**) The dependence of the pneumocytes dynamics for different values on the probability of macrophage infection ρ.

**Figure 2 ijms-23-11364-f002:**
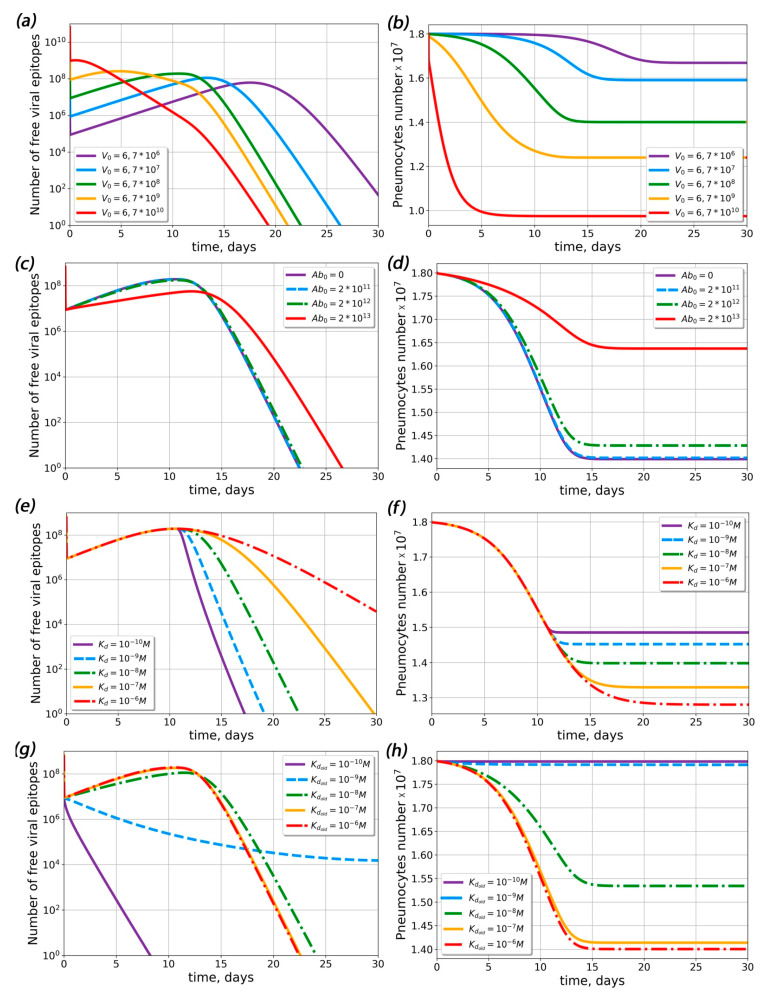
Influence of initial viral load, pre-existing antibody concentration and dissociation constant of the antigen–antibody complex (Kd) on the free viral epitope and pneumocyte dynamics in the complete model. (**a**,**b**) Dynamics of free viral epitope (**a**) and pneumocytes (**b**) at various initial viral loads. (**c**,**d**) Dynamics of free viral epitope (**c**) and pneumocytes (**d**) at different concentrations of pre-existing antibodies. Label *Ab*_0_ denotes the concentration of pre-existing antibodies in lymphoid organs at the onset of disease. (**e**,**f**) Dynamics of free viral epitope (**e**) and pneumocytes (**f**) at various dissociation constants for the antigen–antibody complex for primary antibodies. (**g**,**h**) Dynamics of free viral epitope (**g**) and pneumocytes (**h**) at various dissociation constants for the antigen-antibody complex for pre-existing antibodies. The initial concentration of pre-existing antibodies equaled 10^13^ particles (*Ab_lo_* = 10^13^).

**Figure 3 ijms-23-11364-f003:**
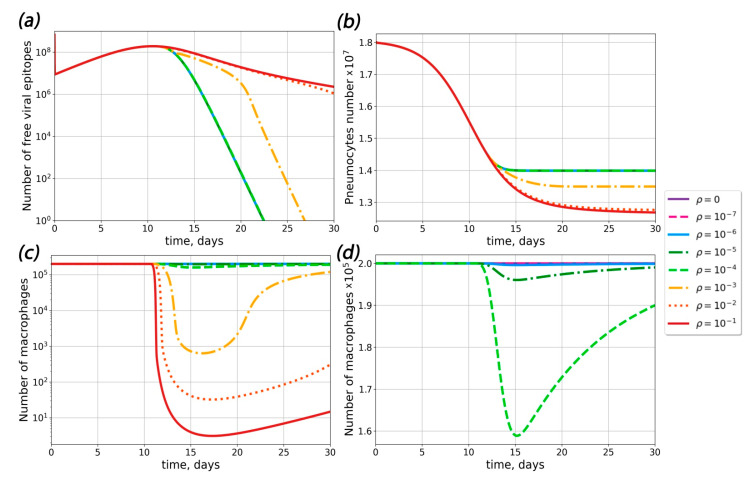
Influence of macrophage infection probability ρ on the course of disease. (**a**) The dependence of the viral response dynamics for different probabilities of macrophage infection ρ. (**b**) The dependence of the pneumocyte dynamics for different probabilities of macrophage infection ρ. (**c**,**d**) The dependence of the macrophage dynamics for different probabilities of macrophage infection ρ.

**Figure 4 ijms-23-11364-f004:**
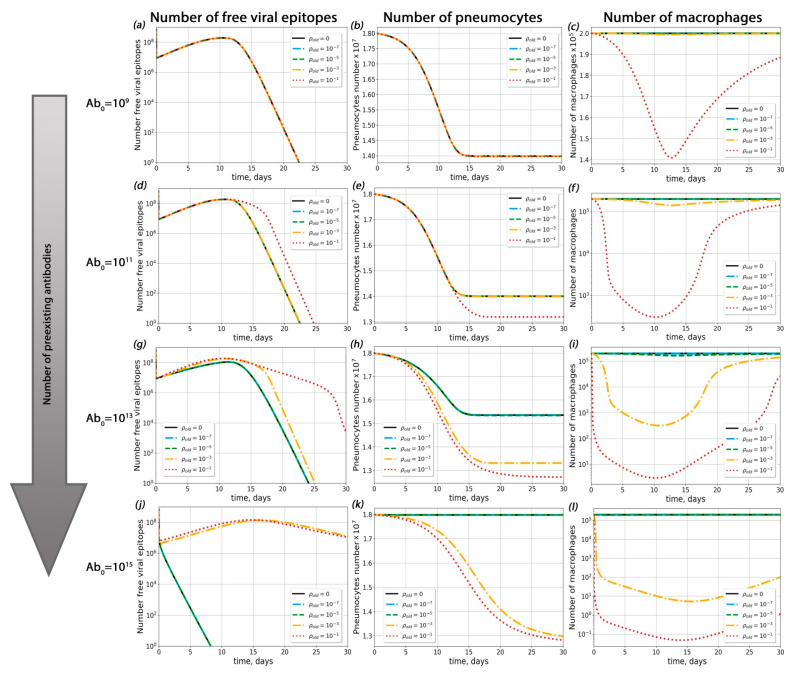
Influence of pre-existing antibodies with different macrophage infection probabilities on possible ADE occurrence. (**a**–**c**) Viral load dynamics (**a**), pneumocyte number (**b**) and macrophage kinetics (**c**) with different macrophage infection probabilities (*Ab_old_* = 10^9^). (**d**–**f**) Viral load dynamics (**d**), pneumocyte number (**e**) and macrophage kinetics (**f**) with different macrophage infection probabilities (*Ab_old_* = 10^11^). (**g**–**i**) Viral load dynamics (**g**), pneumocyte number (**h**) and macrophage kinetics (**i**) with different macrophage infection probabilities (*Ab_old_* = 10^13^). (**j**–**l**) Viral load dynamics (**j**), pneumocyte number (**k**) and macrophage kinetics (**l**) with different macrophage infection probabilities (*Ab_old_* = 10^15^).

**Figure 5 ijms-23-11364-f005:**
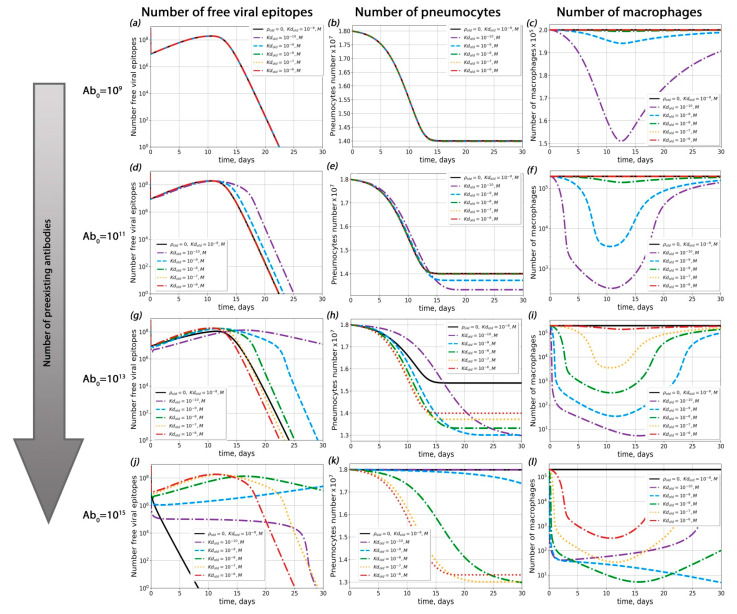
Influence of antigen–pre-existing antibody complex dissociation constant (Kd_old_) on possible ADE occurrence. (**a**–**c**) Viral load dynamics (**a**), pneumocyte number (**b**) and macrophage kinetics (**c**) with different Kd_old_ (*Ab_old_* = 10^9^). (**d**–**f**) Viral load dynamics (**d**), pneumocyte number (**e**) and macrophage kinetics (**f**) with different Kd_old_ (*Ab_old_* = 10^11^). (**g**–**i**) Viral load dynamics (**g**), pneumocyte number (**h**) and macrophage kinetics (**i**) with different Kd_old_ (*Ab_old_* = 10^13^). (**j**–**l**) Viral load dynamics (**j**), pneumocyte number (**k**) and macrophage kinetics (**l**) with different Kd_old_ (*Ab_old_* = 10^15^).

**Figure 6 ijms-23-11364-f006:**
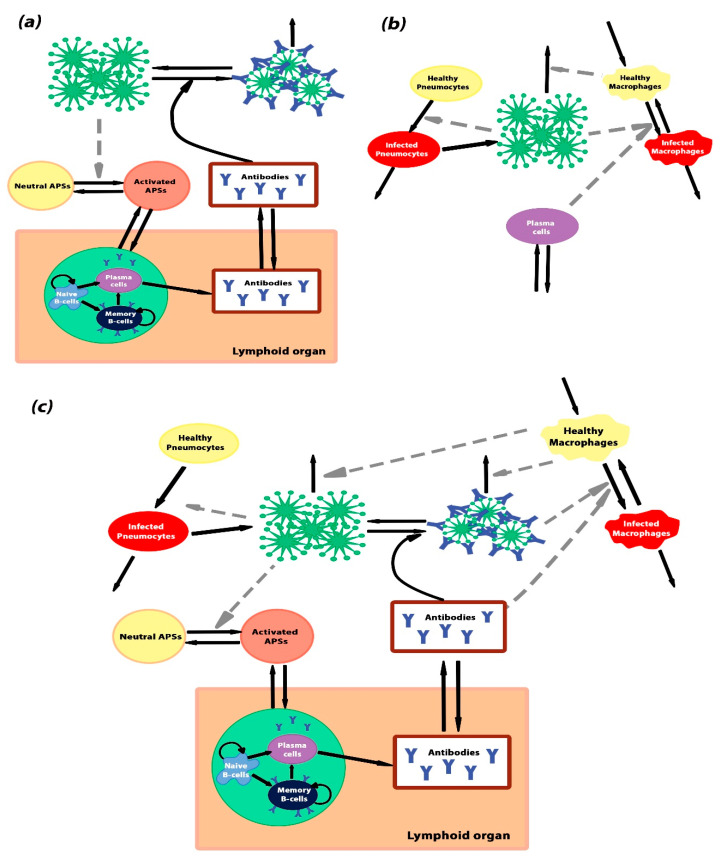
Schematic representations of the models. (**a**) The scheme of model Ab—antibody generation. The model describes the stimulation of the antibody-producing system after an interaction with a pathogen. Activation of B-cells by some antigen-presenting sites (APSs) and their differentiation into antibody-secreting plasma cells and memory B-cells occur in lymphoid organs. The APSs occur in response to the infection. In the absence of pathogens, APSs are in the inactivated state (yellow cycle), and they change the activated state (orange cycle) after infection of the body. (**b**) The scheme of model *Cv*—viral infection of the alveolus. The model describes the processes of pneumocyte and macrophage viral infection and virus degradation through the immune system and thermal degradation. (**c**) The scheme of the complete model, which is the combination of model Ab and model *Cv*. Black solid arrows denote transitions between different states and grey dashed arrows denote indirect interactions.

## Data Availability

The datasets generated and/or analyzed during the current study are available in the Google Drive repository, available online.

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
