# Peer review of "Theoretical Explanation for the Rarity of Antibody-Dependent Enhancement of Infection (ADE) in COVID-19"

_ijms, 2022, doi:10.3390/ijms231911364_

Round 1

Reviewer 1 Report

The work of Boldova et al. is extremely interesting and presents sound results after applying compelling mathematical data and supports results of the present scenario of COVID-19 mass vaccination program in countries of Europa and Americas. My suggestions are listed below:

- Discussion will benefit greatly with the inclusion of the following articles:

Yang X, Zhang X, Zhao X, Yuan M, Zhang K, Dai J, Guan X, Qiu HJ, Li Y. Antibody-Dependent Enhancement: ″Evil″ Antibodies Favorable for Viral Infections. Viruses. 2022 Aug 8;14(8):1739. doi: 10.3390/v14081739.

Danchin A, Pagani-Azizi O, Turinici G, Yahiaoui G. COVID-19 Adaptive Humoral Immunity Models: Weakly Neutralizing Versus Antibody-Disease Enhancement Scenarios. Acta Biotheor. 2022 Aug 13;70(4):23. doi: 10.1007/s10441-022-09447-1.

Hegazy AN, Krönke J, Angermair S, Schwartz S, Weidinger C, Keller U, Treskatsch S, Siegmund B, Schneider T. Anti-SARS-CoV2 antibody-mediated cytokine release syndrome in a patient with acute promyelocytic leukemia. BMC Infect Dis. 2022 Jun 13;22(1):537. doi: 10.1186/s12879-022-07513-0.

Wang S, Wang J, Yu X, Jiang W, Chen S, Wang R, Wang M, Jiao S, Yang Y, Wang W, Chen H, Chen B, Gu C, Liu C, Wang A, Wang M, Li G, Guo C, Liu D, Zhang J, Zhang M, Wang L, Gui X. Antibody-dependent enhancement (ADE) of SARS-CoV-2 pseudoviral infection requires FcγRIIB and virus-antibody complex with bivalent interaction. Commun Biol. 2022 Mar 24;5(1):262. doi: 10.1038/s42003-022-03207-0.

Ajmeriya S, Kumar A, Karmakar S, Rana S, Singh H. Neutralizing Antibodies and Antibody-Dependent Enhancement in COVID-19: A Perspective. J Indian Inst Sci. 2022 Feb 4:1-17. doi: 10.1007/s41745-021-00268-8.

- In my opinion authors should also address ADE in COVID-19 patients treated with convalescent plasma (CP) in the light of the following articles:

Okuya K, Hattori T, Saito T, Takadate Y, Sasaki M, Furuyama W, Marzi A, Ohiro Y, Konno S, Hattori T, Takada A. Multiple Routes of Antibody-Dependent Enhancement of SARS-CoV-2 Infection. Microbiol Spectr. 2022 Apr 27;10(2):e0155321. doi: 10.1128/spectrum.01553-21.

Clark NM, Janaka SK, Hartman W, Stramer S, Goodhue E, Weiss J, Evans DT, Connor JP. Anti-SARS-CoV-2 IgG and IgA antibodies in COVID-19 convalescent plasma do not facilitate antibody-dependent enhance of viral infection. bioRxiv [Preprint]. 2021 Sep 14:2021.09.14.460394. doi: 10.1101/2021.09.14.460394. Update in: PLoS One. 2022 Mar 8;17(3):e0257930.

Dassarma B, Tripathy S, Matsabisa M. Emergence of ancient convalescent plasma (CP) therapy: To manage COVID-19 pandemic. Transfus Clin Biol. 2021 Feb;28(1):123-127. doi: 10.1016/j.tracli.2020.11.004. Epub 2020 Dec 5.

Minor points:

Line 33 – “COronaVirus Infectious Disease-2019 (COVID-19)” – change to “coronavirus Disease 2019 (COVID-19)”

Line 35 – “ [1], [2].” – The format is not correct. Citations of sequential articles should be in the same [], for example [1,2]. Please revise it throughout the text.

Line 41- “ [6]–[10] “ - The format is not correct. Citations of sequential articles should be in the same [], for example [6-10]. Please revise it throughout the text.

Line 48 – “SARS-CoV” – before presenting abbreviation please provide full name in this case: severe acute respiratory syndrome coronavirus (SARS-CoV) – the same logic should be used for MERS-CoV in line 54.

Line 64 – “ SARS-CoV 2” – please change to SARS-Cov-2. Please check the use of “-“ 2 it throughout the text.

Reviewer 2 Report

This theoretical study by Boldova et al. presents a mathematical model of the COVID-19 causing virus (SARS-CoV-2) dynamics in the body. Modeling results explain the previously unexplored question: why there are no reported cases of ADE in COVID-19 patients?  ADE usually develops if immune cells become infected by the virus and start replicating it instead of eliminating. The authors chose human lungs as a system to model and explore conditions that can lead to ADE in COVID-19.  The model describes the main aspects of virus interaction with lung cells and immune system, including antibody production by B-cells, binding of antibodies to virus, elimination of virus-antibody complexes by macrophages, infection of lung cells (pneumocytes) and their production of viral particles, infection of macrophages.  The model parameters are experimentally validated: partially they are estimated by fitting to experimental data from published research on COVID-19 patients and partially are taken from previously published theoretical papers.  Thus, this study provides a wisely constructed detailed model which is a valuable tool for the field of infectious diseases and for study of COVID-19 in particular. It allows exploration of the viral infection course and outcome and for estimating an effect of different parameters on the course of disease.

The results of the computations show how viral load and pneumocytes survival depend on various parameters such as initial antibody concentration (due to vaccination or previous infection), affinity of antibodies to the virus, initial viral dose, probability of the macrophage infection.  From these computations, authors draw the conclusion that ADE in COVID-19 cannot be distinguished from the other severe cases of disease caused by low antibody concentration and hence high cell death rate.  Although this qualitative conclusion is somewhat supported by the modeling results, it raises several questions about model construction, which need to be addressed by explanations in the text or by refining the model.  

Overall, the paper is mostly well written. However, it requires a number of improvements to make the main aspects of the model clear to readers and to substantiate main conclusions. It also requires editing because many typos and inappropriate use of words make it hard to read.  Please see below the list of suggested changes:

Main suggestions:

1)    In the introduction it is mentioned that the cause of immune cells’ infection in ADE is “insufficient binding” of antibodies to virus particles (lines 44-45). However, in the equation #14 that describes M*, the macrophage infection rate is proportional to antibody-virus complexes concentration and doesn’t contain any term which depends on antibody affinity. Infection probability, rho, doesn’t depend on Kd either. Consequently, higher Ab affinity leads to higher macrophage infection in the model together with increased pneumocytes survival, although in reality maximum macrophages infection rate should be at some suboptimal antibody affinity.   

Please explain in the text why equation #14 and corresponding “Complete model”  equations don’t account for “insufficient binding” mechanism. Alternatively, take into account this mechanism in the model and show how will results differ if macrophage infection rate will maximize at some intermediate Kd.

2)    What is considered as ADE state in the output of computations? How does it correspond to the clinical "definition" of ADE, what parameters are measured in patients to detect ADE?  Indicate in the text and on the Fig.4, which regions of graphs correspond to ADE – is it a drastic death of pneumocytes or no drop in viral load?  Or add a combined plot, where ADE outcome can be easily directly compared to similar outcomes with no macrophage infection.
Arguing in conclusion that ADE is indistinguishable from the other courses of disease with no macrophage death, indicate which scenarios on the graphs are referred to.  Also it is suggested to move some of the plots showing macrophage numbers from Supp.Info to the main Figures to show the dynamics of macrophages alongside with pneumocyte numbers and viral load (such as in Fig.S7)

3)    What would be the production rate of viral particles by infected macrophages compared to production by pneumocyte? Explain in the text why equation #17 doesn’t contain the term describing viral production by infected macrophages (which would stimulate the occurrence of ADE). According to f.e. Fig. S5C as many as ~10^5 macrophages acquire infected state and in principle can significantly contribute to viral production, making ADE scenario more severe.

Minor comments:

1)    The main conclusion of the paper states that only pre-existing antibodies can contribute to ADE.  Can the “Model-Ab” of antibody production be completely neglected in this case? Or do newly produced antibodies also contribute to some extent to the outcome of the model?  Add this information to the discussion.

2)    Move some of the plots showing macrophage numbers to the main Figures.

3)    For Fig.3 it is suggested to present plot 2D plots for few Ab0 values to ease direct comparison, and move 3D plots to Supp.Info.

4)    What is time =0 in the graphs? In experimental data used for fitting 0 is time of onset of symptoms. Indicate that in the graphs.

Textual changes:

1)    To make the paper accessible to broader readership I would suggest to include meaning of abbreviations and special terms at the place of first mentioning and/or add glossary. For example: APS, inoculum dose, seroconvertion, HAE

2)    Use only one term for the same notion: f.e. Inoculum dose = initial viral load.

3)    Line 78: in the absence _of_

4)    Line 114: possibilities _of_

5)    Line 134: _on_ the course..

6)    Line 143: delete “impact”

7)    Line 151: noticeably

8)    Lines 179-182: not clear what do these 2 sentences with repeating words mean

9)    Line 185: “dependent on” instead of “determined by”

10) Line 197: dynamics

11) Line 212: “As it was found” in previous studies?  Then “As it was already shown”..; ”In accordance with previous studies”

12)   Line 214: duration of the disease became _shorter_.

13) Line 220-221: why is there a reference?  The sentence should then include information from the reference , f.e. “as shown in this study”

14) Line 223-224: Is it a downside of this model that it is not in accordance with other models which are not in accordance with experimental data?  Clarify this sentence.   

15) Line 243: “decline” instead of “deterioration”

16)  Line: 248-249: Sentence is unclear. What is shown in refs 37,38? Why it can be assumed that vaccination doesn’t cause ADE if the paper only shows that positive effect from pre-existing antibodies outcompetes ADE effect, so that disease outcome is in the end better than without  pre-existing antibodies.

17) Line 285: “Vf” not “V”

18) Line 296: Ab-lung

19) Line 299: “describe” instead of “concern”

20) Line 310: “cannot affect..”

21) Line 311: dynamicS

22) Line 332: “occurs” instead of “becomes”

23) Line 372: events

24) Equation for alpha_c : explain why this function is chosen, why is +10?
